# Defogging Algorithm Based on Polarization Characteristics and Atmospheric Transmission Model

**DOI:** 10.3390/s22218132

**Published:** 2022-10-24

**Authors:** Feng Ling, Yan Zhang, Zhiguang Shi, Jinghua Zhang, Yu Zhang, Yi Zhang

**Affiliations:** National Key Laboratory of Science and Technology on Automatic Target Recognition, College of Electronic Science and Technology, National University of Defense Technology, Changsha 410073, China

**Keywords:** polarization characteristics, image defogging, image defogging, atmospheric transmission model

## Abstract

We propose a polarized image defogging algorithm according to the sky segmentation results and transmission map optimization. Firstly, we propose a joint sky segmentation method based on scene polarization information, gradient information and light intensity information. This method can effectively segment the sky region and accurately estimate the global parameters such as atmospheric polarization degree and atmospheric light intensity at infinite distance. Then, the Gaussian filter is used to solve the light intensity map of the target, and the information of the polarization degree of the target is solved. Finally, based on the segmented sky region, a three-step transmission optimization method is proposed, which can effectively suppress the halo effect in the reconstructed image of large area sky region. Experimental results shows that defogging has a big improvement in the average gradient of the image and the grayscale standard deviation. Therefore, the proposed algorithm provides strong defogging and can improve the optical imaging quality in foggy scenes by restoring fog-free images.

## 1. Introduction

The imaging quality of natural scenes depends on weather conditions, such as haze and fog. In a foggy scene, water vapor in the atmosphere is stable owing to suspended water droplets. Owing to the strong scattering properties of the suspended particles, air light is scattered and absorbed during propagation, reducing visibility [1,2]. Consequently, optical imaging of foggy scenes presents low visibility, low contrast, blurry details, and even color distortion [3], compromising the image quality for tasks like target detection and recognition. To improve the accuracy and reliability of such tasks, image defogging and clear imaging under foggy conditions are extremely valuable in practice.

Research on image defogging has been actively conducted in recent years. Most defogging algorithms are based on direct image processing or considering the foundation of atmospheric transmission models. Defogging based on image processing such as Retinex [4,5], wavelet transform [6,7,8,9] and homomorphic filtering [10,11]. These methods aim to improve the image quality through the global or local image enhancement, and do not take the physical causes of image degradation on foggy days into account. Defogging algorithms based on atmospheric transmission models determine image contrast degradation caused by the underlying physical mechanisms. By applying the physical model to a foggy image, the fog-free image can be restored. This type of algorithm focuses on image degradation owing to fog, in which light is scattered and absorbed during propagation by suspended water particles in the atmosphere. A representative algorithm is a dark channel prior proposed by He et al. [12]. They analyzed massive experimental data captured over several years in fog-free weather, finding that at least one of the three image channels in the non-sky region shows an extremely low pixel value level that is assumed to be infinitely close to zero. According to this prior knowledge and an atmospheric transmission model, they estimate the transmission map and reconstruct fog-free images. Other image dehazing algorithms based on atmospheric transmission model, such as color attenuation prior algorithm and optimized contrast enhancement algorithm, both have a good performance in the field of single image dehazing.

In recent years, with the rapid progress of convolutional neural network (CNN), image dehazing algorithms based on neural networks have been developed successively. CAI first proposed the DehazeNet for image dehazing [13], and due to the emergence of GAN, CHEN proposed an end-to-end image dehazing algorithm [14], which called GCANet. This method uses smooth extended convolution instead of extended convolution, which solves the problem of grid artifacts, and improves the image defogging effect.

Polarimetric dehazing method was first proposed by Schechner in 2001 [15,16]. They acquired the polarimetric information by obtaining two orthogonal polarimetric images, and the model parameters are estimated to remove haze. Zhang proposed an image fusion method based on the correlation and information complementarity of different polarimetric images, which has a good effect on image restoration under mist weather conditions [17]. Zhou proposed an image fusion dehazing algorithm based on multi-scale singular value decomposition to overcome the lack of robustness of traditional polarimetric dehazing algorithms [18]. Huo proposed a method which introduces polarimetric information into the traditional dark channel prior to enhance the identification degree between different objects [19]. The above study shows that the polarimetric dehazing method is effective in many different hazy environments. However, when processing some images with high scene light intensity and large area of sky, the traditional image dehazing algorithm based on polarimetric information is easy to appear halo effect in the restoration image, which has a serious impact on the quality of the reconstructed fog-free image.

To address the shortcomings of existing methods, we propose a defogging algorithm according to the sky segmentation results and transmission map optimization. First, the polarimetric characteristics are extracted from polarimetric images and the angle of polarization (AOP). Then, the sky region is jointly segmented using the AOP and image of scene intensity (IOI) to obtain the polarization degree of air light ($P_{a}$) and air-light intensity at infinite distance ($A_{\inf}$) relative to the sky region. In addition, Gaussian blur filtering is used to estimate the air-light intensity image. Then, the object light intensity image and object polarization degree ($P_{d}$) are obtained to construct an initial transmission map. The map is refined with the sky region and guided image filtering, finally providing a fog-free image. We use different methods to deal the hazy images under different scenarios, in order to test their dehazing ability compared to the proposed methods. Furthermore, the average gradient, gray variance and no reference image quality evaluation are used here to assess the quality belongs to different method’s result. The precision of the experimental analysis shows that the proposed method can effectively solve the halo phenomenon caused by the light intensity in the hazy images, and effectively improve the quality of the reconstructed images. Furthermore, the proposed algorithm also has good experimental results in low light intensity scenes.

## 2. Atmospheric Transmission Model

The two primary causes of image degradation in foggy scenes are air-light scattering into stray light during propagation, which results in image blur, and absorption and scattering of the reflected light by water particles suspended in air, which reduce image brightness and contrast [20,21,22,23].

In polarized imaging, when light scatters during propagation, its polarization changes accordingly, and the air light changes from non-polarized to partially polarized. The polarization of an object depends on both the scattering effect caused by water particles suspended in air and the material and surface roughness of objects. Therefore, polarization information of the scene acquired through optical imaging includes the polarization information of both air light and objects [24]. The atmospheric transmission model used in this study is illustrated in Figure 1.

Through the atmospheric transmission model, the light reaching the optical imaging system can be expressed as:(1)$$I=Lt+A_{\inf}(1-t)$$
where I represents the intensity of the scene, L represents the fog-free image, $A_{\inf}$ represents air-light intensity at infinite distance, and t represents the transmission map. By including the polarization degree, we obtain:(2)$$I_{d}=IP=DP_{d}+AP_{a}$$
where $P_{}$, $P_{d}$, and $P_{a}$ are the polarization degrees of the scene, object, and atmosphere, respectively, D and A are the light intensities of the objects and air light, respectively, and $I_{d}$ is the polarization-difference image.

By letting:(3)I=D+A

A and D can be expressed using polarization parameters as follows:(4)$$\begin{array}{l} D=\frac{I(P-P_{a})}{P_{d}-P_{a}}\\ A=\frac{I(P-P_{d})}{P_{a}-P_{d}} \end{array}$$

Based on the atmospheric transmission model and (2)–(4), transmission map t can be expressed in terms of $P_{d}$ as:(5)$$t_{\mathrm{inital}}=1-\frac{P_{d}I-I_{d}}{(P_{d}-P_{a})A_{\inf}}$$
which is a function of $I_{d}$, $P_{d}$, the light intensity of images I, $A_{\inf}$, and $P_{a}$. Therefore, the transmission map t can be obtained by calculating the above-mentioned parameters.

## 3. Stokes Vector and Polarized Images

### 3.1. Extraction of Scene Polarization Characteristics

Polarized light can have trigonometric, Jones vector, Stokes vector, or graphic representation. The Stokes vector representation is widely used given its detection convenience and intuitive expression [25]. Thus, we use this representation to describe polarization characteristics.

The Stokes vector representation is a matrix representation of polarization with four rows and one column, where the four parameters describe the intensity of different characteristics:(6)$$S={\left[S_{0},S_{1},S_{2},S_{3}\right]}^{T}$$
where $S_{0}$ is the total light intensity of the scene, $S_{1}$ is the difference in intensity between the linear polarized horizontal and vertical components, $S_{2}$ is the difference in intensity between the linear polarized components captured at 45° and 135°, and $S_{3}$ is the difference in intensity between the left- and right-handed polarized light components. In general, circularly polarized light is negligible in polarized optical imaging. The images captured by a four-angle polarization detector are denoted as $I_{0}$, $I_{45}$, $I_{90}$, and $I_{135}$. Each parameter in the Stokes vector can be expressed as:(7)$$\begin{array}{l} S_{0}=I_{0}+I_{90}\\ S_{1}=I_{0}-I_{90}\\ S_{2}=I_{45}-I_{135} \end{array}$$

For the Stokes vector, the degree of linear polarization (DOP) and AOP can be expressed as follows:(8)$$\left\{\begin{array}{l} Dop=\frac{\sqrt{S_{1}^{2}+S_{2}^{2}}}{S_{0}}\\ \theta =\frac{1}{2}\tan^{-1}(\frac{S_{1}}{S_{2}}) \end{array}\right.$$

### 3.2. Joint Segmentation of Sky Region

He et al. considered that the intensity at one of the RGB (red–green–blue) image channels is always close to 0 in their dark channel prior algorithm. According to this assumption, the dark channel image was obtained and $A_{\inf}$ was estimated. The dark channel prior algorithm avoids errors related to the heuristic selection of $A_{\inf}$. However, the sky region does not follow the dark channel prior assumption. When a large sky region is acquired in an image, the corresponding $A_{\inf}$ is inaccurate, producing texture and blocking effect in the restored fog-free image, which reduces the image quality. In the literature [15,16], a small region far enough from the target was selected as the sky region, and the pixel in this region was calculated and solved to obtain the $A_{\inf}$. However, this method has certain subjectivity and cannot ensure that the selected sky region is globally representative.

To prevent this problem, we perform automatic sky region segmentation based on AOP and IOI to accurately estimate $P_{a}$ and $A_{\inf}$ from a correctly segmented sky region.

In a foggy environment, the polarization angles of the scattered light and the reflected light are generally different, and the polarization angles of the reflected light of different materials are different. Owing to the uniform atmospheric distribution in the sky region, the change in polarization angle in the sky region is relatively gentle, while the angle of polarization transforms at the junction of the sky and the target region will jump. According to the atmospheric transmission model, with an increase in the transmission distance, the light intensity of the atmospheric scattered light increases while the light intensity of the target reflected light decreases. Since the transmitted light in the sky region is approximately infinity, the light intensity in the sky region must be higher than that in the target region. Therefore, an automatic sky region segmentation algorithm combining polarization angle image and scene light intensity image is proposed in this paper. The segmented sky region is used to accurately estimate the atmospheric polarization degree and the atmospheric light intensity at infinity.

Sobel operator is a common edge detection operator [26,27,28]. Sobel operator calculates the gradient of the corresponding region of the image. Since the gradient information of the region reflects the degree of change of the image in the region, the gray value at the junction of the target and the sky will change obviously, and its gradient value will also change accordingly. In the sky region and the target region, the gray value changes relatively smoothly, and the corresponding gradient changes are small; hence, the sky region can be segmented theoretically.

Starting from the pixels in the second row and second column of the upper left corner of the polarization angle image and scene light intensity image, the image gradient of polarization angle and the scene light intensity image were successively calculated to obtain the image gradient map of the polarization angle and scene light intensity image. Because the atmosphere in the sky region has the characteristics of uniform distribution, and the intensity value of the target changes relatively gently, if the gradient jump at a point in the scene light intensity image is less than the preset threshold, we think it is not at the edge of the target and the sky. In order to ensure the authenticity, the gradient jump of the same pixel position in the gradient map of the polarization Angle image is judged again whether it is greater than the preset gradient value. If the gradient jump of this point is greater than the preset threshold, it is considered that this point may be noise or edge point in the scene light intensity image, and the value of this point is taken as 0. Otherwise, if the gradient jump of this point is smaller than the preset threshold at the same time, this point is considered a non-edge point. In the sky area and the target area, the gradient will not be greater than the preset threshold because the pixel value changes smoothly. In this case, the target light intensity is introduced to judge the area where the pixel belongs, if the pixel is judged as a non-edge point. If the pixel value of the corresponding pixel in the scene light intensity image is greater than the preset threshold, the point is considered as the sky area, and the value is taken as 1; otherwise, it is 0. Finally, the maximum area connectivity is used to connect the target area and remove the noise points in the sky area.

By comparing the coincidence degree between the real scene of the image and the extracted sky area, the extraction effect is judged. When the polarization angle gradient threshold is 60, the scene light intensity gradient threshold is 60, and the scene light intensity threshold is 180. The extraction results for the sky region are shown in Figure 2a. Gradually increase the value of the three thresholds, get b–g. When the polarization angle gradient threshold is 70, the scene light intensity gradient threshold is 95, and the scene light intensity threshold is 205, the sky region can be extracted. However, because the spectral intensity in the middle of the target is close to the light intensity in the sky region, this part is wrongly extracted as the sky region. By further increasing the threshold of scene light intensity, the sky region and the target can be completely segmented. However, some sky region at the target is not infinite, and the intensity characteristics of this region are closer to those of the targets on both sides of it, so this part is segmented into the target region. If the threshold of scene light intensity continues to increase, the image of the miss-elected area will increase rapidly, which will cause interference to the subsequent transmission optimization and restoration of the fog-free image.

$P_{a}$ and $A_{\inf}$ are important parameters for defogging because they describe air light. They are generally considered global parameters owing to the smooth color distribution of the air in images. We estimate $P_{a}$ and $A_{\inf}$ using the sky binary template, DOP, and IOI as follows.

The image of scene polarization degree in the sky region is obtained by multiplying the sky binary template and DOP. Then, according to the statistics of the polarization degree corresponding to each pixel in the sky region, the most frequent value is taken as $P_{a}$. To estimate $A_{\inf}$, the sky binary template is multiplied by the scene-light intensity image to obtain an image that preserves only the sky region. The average of the 100 largest pixel values is taken as $A_{\inf}$. The estimation error of *P_a_* and *A*_inf_ caused by interferences in the sky region is prevented using statistics.

### 3.3. Calculation of Polarized-Difference Image

Two perpendicularly polarized images are used to measure the polarization degree. Their difference is denoted as $I_{d}$, which is important for restoring a fog-free image.

To obtain polarized-difference images, the artificial rotating polarizer or fixed-angle shooting method is used to obtain dual-angle images [29] and estimate the differential images. However, as objects in a scene show diverse polarization characteristics, these methods cannot provide the best Id image by only relying on orthogonal imaging. Hence, we propose a method to directly obtain $I_{d}$ by combining the Stokes vector and polarization degree of the scene.

The polarization degree and light intensity of the scene can be expressed by dual-angle images as follows:(9)$$P=\frac{I_{worst}-I_{best}}{I_{worst}+I_{best}}=\frac{I_{d}}{S_{0}}$$

As P is estimated by the Stokes vector during polarization characteristic extraction (Section 3.1), $I_{d}$ can be expressed as:(10)$$I_{d}=S_{0}P$$

Thus, image $I_{d}$ can be obtained directly from P and $S_{0}$, avoiding calculations of dual-angle images and reducing the algorithm complexity and running time.

## 4. Calculation of Air-Light Intensity Image Based on Gaussian Blur Filtering

### Calculation of Light Intensities A and D

The most common methods to estimate the image of intensity A are the dark channel prior and median filtering [30]. As mentioned above, the dark channel prior is not suitable for scenes with large sky regions because color can be easily distorted, and the block effect may appear in restored images. On the other hand, median filtering fails to preserve image edges, likely leading to the loss of details after the restoration of fog-free images. Instead, we use the Gaussian blur algorithm to estimate the air-light intensity image and ensure smooth filtering while maintaining the image edges.

Gaussian blur performs filtering by assigning weights according to a normal distribution. Owing to the continuity of image pixels, nearer pixels have higher correlation, rendering normal distribution weighting suitable for filtering. The position of the center of a filtering window is set as the primary pixel, and the weights of other pixels in the window are allocated according to the position in the normal distribution. Thus, the weighted average can be obtained as a filtered value of the center. A two-dimensional Gaussian function can be expressed as:(11)$$G(x,y)=\frac{1}{2\pi \delta^{2}}e^{-(x^{2}+y^{2})/2\delta^{2}}$$

The filtered image *I* is thus given by
(12)$$I=\sum_{x,y\in \Omega }G(x,y)\cdot A(x,y)$$
where *σ* is the size of the weight matrix, *A* represents the image to be filtered, and Ω is the window area.

We adopt the method in the reference [30] for differential filtering. The estimated polarized air-light images are obtained from the original polarized images: $I_{0}$, $I_{45}$, $I_{90}$, and $I_{135}$, according to the Formula (13):(13)$$\begin{array}{l} B=Gauss(I)-Gauss(\left|I-Guass(I)\right|)\\ A=\max(\min(aB,I),0) \end{array}$$

The object light intensity at each angle is calculated according to Formula (3), and the polarization degree of the target can be solved by Stokes vector. Examples of polarized air-light images, polarized object images at different angles, the light intensity of the air-light, the light intensity of the object and polarization degree of the target are shown in Figure 3 below:

## 5. Estimation of Transmission Map t and Restoration of a Fog-Free Image

Through the parameters above all to estimate the transmittance map t.

Due to the use of filter method are applied to solve the target light intensity images, the result is just an approximation of the target light intensity map, so we use the above parameters for the estimation of the transmittance map t is usually associated with errors, so we need to optimize the transmission rate, ensure that the quality of the fog-free scene finally restored.

Here, we propose a three steps method for transmission map optimization in this study:

Step 1: Use statistics to calculate the pixel value of the sky area. If the pixel value of the sky area is greater than 50%, the scene is considered a large-area sky scene, and the sky needs to be optimized to avoid the halo effect caused by the large-area sky area. The sky region was used to optimize the initial transmittance of the solution, and the pixel value of the sky region in the transmittance map was randomly assigned between 0.01 and 0.1 to ensure that the transmittance of the sky region was close to 0, so that there was no large area halo interference in the recovered fog-free scene image.

Step 2: Use guided filtering [31] to optimize the results of Step 1 to ensure the texture and details of the transmittance in the target area.

Step 3: If the image does not belong to a large sky area, direct guided filtering is used to optimize the initial transmittance results.

The transmittance map is obtained as below Figure 4:(14)$$\left\{\begin{array}{l} t(x,y{)}_{mid}=\left\{\begin{array}{cc} \min(t{\left(x,y\right)}_{inital},R(0.01,0.1)) & \left(x,y\right)\in \Omega_{s}\\ t(x,y) & \left(x,y\right)\in \Omega_{o} \end{array}\right.\\ t{\left(x,y\right)}_{final}=G((t{\left(x,y\right)}_{mid})) \end{array}\right.$$

In the above formula, $R_{}$ stands for the random assignment. $\Omega_{s}$ represents the sky region, $\Omega_{o}$ represents the object region, and $G_{}$ represents the guided filtering.

Using (1), the fog-free restored image L can be obtained as:(15)$$L=\frac{D}{t_{final}}$$

Examples of the hazy image I and restored fog-free image L are shown in Figure 5.

In order to better show the process of the algorithm in this paper, the flowchart is shown in the following Figure 6:

## 6. Experimental Analysis

To verify the defogging effect of the proposed method, we compared it with the dark channel prior and the single image dehazing using a multilayer perceptron (SIDMP) [32], which was proposed by Colores in recent years, in foggy scenario, obtaining the results shown in Figure 7.

Figure 7 scenario1–scenario3 shows different outdoor experimental scenes. In Figure 7a is the original fog-containing scene, Figure 7b is the dark channel processing result, Figure 7c is the processing result of the SIDMP, and Figure 7d is the defogging result of the proposed method. It can be seen that all the three dehazing algorithms have obvious dehazing effect. However, when dealing with high light intensity and large area sky image, large area halo effect will appear in the non-hazing image reconstructed by dark channel dehazing algorithm, which reduces the image quality and affects the observation of image details. The SIDMP reduces the brightness of the halo, but cannot effectively suppress the area of the halo, and the target is still in a large area of the halo background. When the fog-free image is reconstructed using the method proposed in this paper, the area of halo is effectively reduced, and the visual quality of the image is improved by optimizing the sky area.

Figure 8 is the local enlarged image corresponding to Figure 7. Based on the analysis of Figure 7, it can be seen that the proposed algorithm can better recover the target information and realize the reconstruction of foggy scenes. At the same time, compared with the dark channel algorithm, the proposed algorithm can effectively improve the detail contrast of the reconstructed image and realize the restoration of the target detail texture.

The gray histogram statistics of the enlarged details in scene 2 are performed, and the results are as follows Figure 9:

Through the gray histogram distribution, we can find that the gray value of the original fog image is mainly distributed in the range of 60–200, and there are a lot of gray value distribution in the high brightness area, which is close to 200. Dark channel method and SIDMP do not effectively realize the extension of image gray distribution. The method proposed in this paper extends the gray distribution to the interval of 10–180, and effectively reduces the gray value distribution in the highlighted area, which means, it effectively reduces the influence of atmospheric light on the image.

To quantitatively evaluate the effectiveness of the proposed algorithm, we obtained the average gradient and grayscale variance of the original foggy images and fog-free images obtained by different algorithms in the scenarios shown in Figure 7.

The average gradient of an image represents the change rate of its grayscale level, which can be used to indicate the sharpness of an image. A larger average gradient indicates higher sharpness, being an important indicator of the expression ability of image details.
(16)$$\bar{G}=\frac{1}{\left(m-1)(n-1\right)}\sum_{i=1}^{m-1}\sum_{j=1}^{n-1}\sqrt{\frac{{\left(F(i,j)-F(i+1,j)\right)}^{2}+{\left(F(i,j)-F(i,j+1)\right)}^{2}}{2}}$$
where F(i,j) represents the gray value of the pixel at position (i,j) in the image and $\overset{-}{G}$ is the average gradient of the image. in the image.

The grayscale variance indicates the dispersion of the distribution of grayscale values. A larger grayscale variance indicates a more scattered distribution of grayscale values:(17)$$std=\sqrt{\frac{1}{mn}\sum_{i=1}^{m}\sum_{j=1}^{n}{\left(F(i,j)-u\right)}^{2}}$$
where mn is the total number of pixels in the image, F(i,j) gray value of the pixel at position (i,j) in the image, and μ is the value of the gray mean.

By analyzing the data quantification results in the Table 1, it can be seen that the statistical characteristics of the data of the defogging and restoring images by the method proposed in this paper are greatly improved compared with the processing results of the original fog-containing scene. Compared with the dark channel de-hazing algorithm, when processing the data of scenario1–scenario3, the average gradient of the result come from the proposed method is better because the halo phenomenon in the restored image is effectively suppressed. That is, the clarity and detail expression ability of the restored image is stronger than that of the dark channel algorithm. In terms of gray standard deviation, since there are halos in scenario1–scenario3, the gray variance value of the dark channel processing result is larger than that of the proposed algorithm, which also proves that the proposed method can effectively suppress the halo effect. The quantitative analysis results obtained by SIDMP are similar to those obtained by the dark channel method.

To test the ability of proposed method in the low light intensity scenarios, here we give another tow experimental results, and another dehazing method [33] proposed in the past two years, which is called Dual Transmission Maps Strategy (DTMS), to compare the results Figure 10:

Through the visualization results, we can find that the algorithm in this paper can also achieve a better dehazing effect when dealing with such low light intensity scenarios.

The gray histogram statistics of the scenario4 are performed, and the results are as follows Figure 11:

Through the gray histogram visualization of scenario4, it can be seen that the method proposed in this paper can effectively improve the gray histogram distribution of images, enhance the images contrast and improve quality of reconstructed fog-free images.

In order to effectively evaluate the quality of reconstructed fog-free image and fog-containing image, the non-reference quality evaluation index, which is called Natural Image Quality Evaluator (NIQE) [34], is adopted to evaluate the image data of scenario4 and scenario5. NIQE is an objective evaluation index that extracts features from natural landscapes to test images, a lower NIQE value corresponds to a higher overall naturalness, representing a higher image quality. The results are shown in Table 2.

According to various evaluation results without reference, in scenario4 and scenario5, the dehazing effect of the proposed method is improved compared with the DTMS.

## 7. Conclusions

In order to solve the halo phenomenon in the dehazing process of high light intensity image with a large region of sky, a joint sky region segmentation method and a transmission map optimization method based on polarimetric information are proposed in this paper. The idea of the proposed method in this paper is that the air light and object light can be separated by Gaussian blur, based on the different frequency characteristic. Then, the image dehazing reconstruction is completed by calculating the parameters in the dehazing model.

The accurate segmentation of sky region plays an important role in the process of image dehazing. The estimation of important parameters such as $P_{a}$ and $A_{\inf}$ can be realized by accurate sky segmentation results. Compared with the traditional method of manually selecting sky area and estimating parameters, the method of parameter estimation combined with sky region is more accurate and reliable. At the same time, the sky region in the transmission map is optimized by using the sky segmentation results. By adjusting the atmospheric transmittance in the sky region, the halo phenomenon in the reconstructed image can be effectively reduced, the visual observation effect and image quality can be better improved. The experimental results show that the proposed method can effectively dehaze and reconstruct the high light intensity images with a large region of sky, greatly reduce the halo effect in the reconstructed images. Combined with the experimental results of scenario4 and scenario5, it is proved that the proposed method also has a good defogging effect when processing low light intensity image. It is proved that the proposed method can achieve good defogging effect under various scenarios.

However, through the experimental results, it can be found that the proposed algorithm is easy to cause a certain degree of color distortion in the process of dehazing reconstruction, especially when there are trees and other objects in the scene. Through the analysis of the experiment, the reason for this phenomenon may be that only approximate results can be obtained when Gaussian blur is used to separate air light from the scene, but the complete and accurate separation cannot be achieved. The next work will focus on optimizing the separation method in order to get better results.

## Figures and Tables

**Figure 1 sensors-22-08132-f001:**
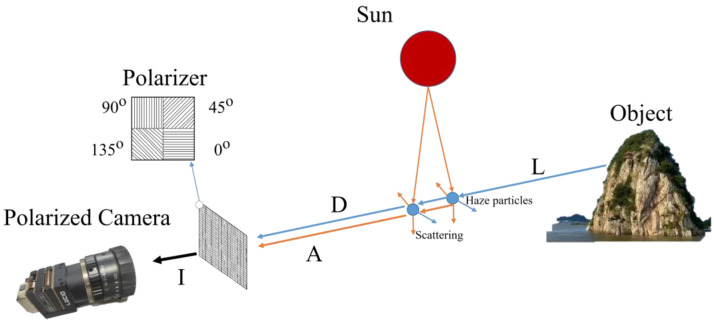
Atmospheric transmission model.

**Figure 2 sensors-22-08132-f002:**
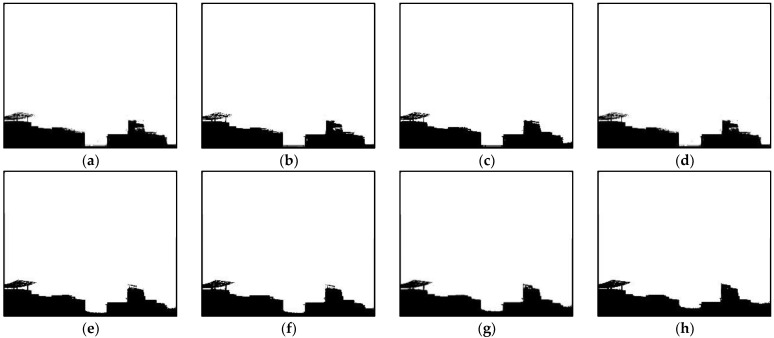
Segmentation example of sky region, (**a**–**h**) show the different results of the sky segmentation under different parameters, (**a**) th1 = 60, th2 = 60, th3 = 180, (**b**) th1 = 70, th2 = 70, th3 = 190, (**c**) th1 = 70, th2 = 80, th3 = 190, (**d**) th1 = 70, th2 = 80, th3 = 200, (**e**) th1 = 70, th2 = 90, th3 = 200, (**f**) th1 = 70, th2 = 95, th3 = 200, (**g**) th1 = 70, th2 = 95, th3 = 205, (**h**) th1 = 70, th2 = 95, th3 = 215.

**Figure 3 sensors-22-08132-f003:**
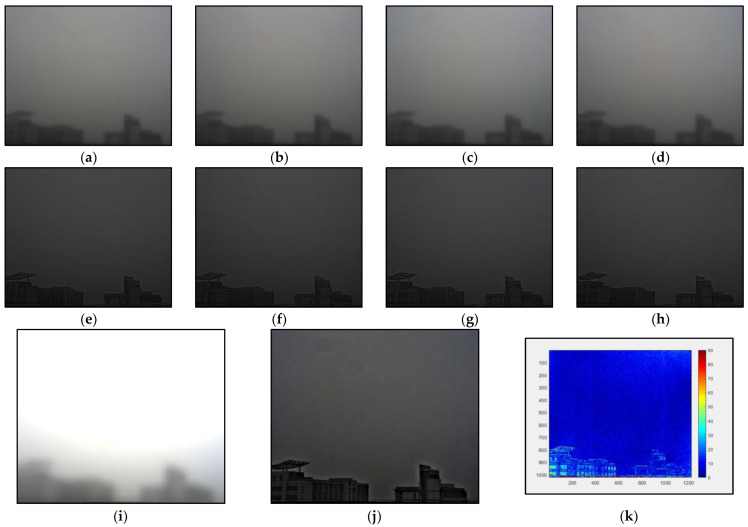
Result of the Gaussian blur filtering, (**a**–**d**) the polarized air-light images at fours angle, (**e**–**h**) the polarized object images at fours angles, (**i**) the light intensity of the air-light, (**j**) the light intensity of the object, (**k**) DOP of the object.

**Figure 4 sensors-22-08132-f004:**
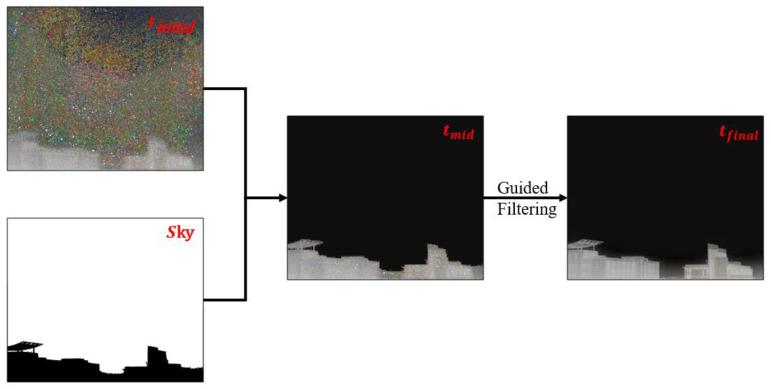
The optimization process of the transmittance map.

**Figure 5 sensors-22-08132-f005:**
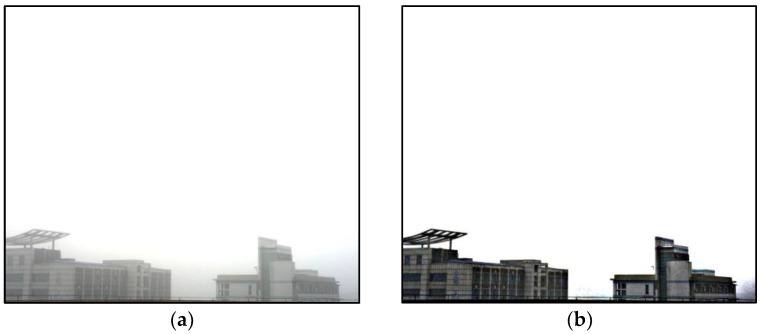
The comparison between original hazy image and dehazed image, (**a**) original hazy image, (**b**) dehazed image.

**Figure 6 sensors-22-08132-f006:**
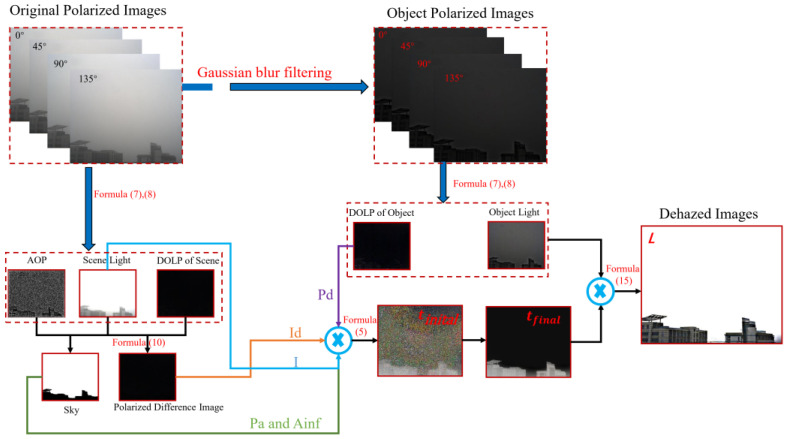
Algorithm flow chart.

**Figure 7 sensors-22-08132-f007:**
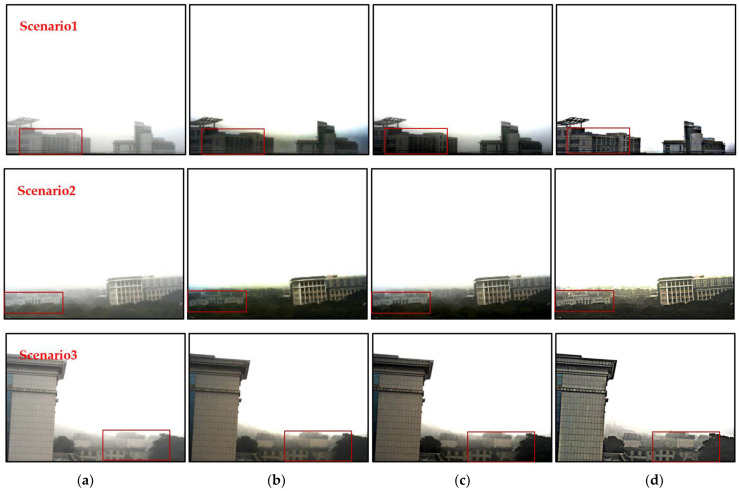
The comparison between different methods, (**a**) the original hazy image, (**b**) the result of dark channel, (**c**) the result of SIDMP, (**d**) the result of proposed method.

**Figure 8 sensors-22-08132-f008:**
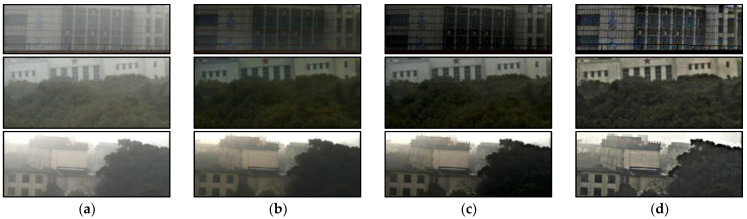
The comparison of the enlarged details, (**a**) the original image, (**b**) the result of dark channel, (**c**) the result of SIDMP, (**d**) the result of proposed.

**Figure 9 sensors-22-08132-f009:**
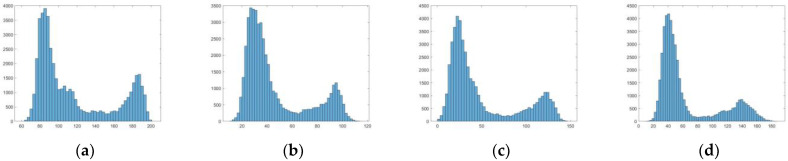
Histogram comparison results, (**a**) result of the original hazy images, (**b**) result of dark channel, (**c**) result of SIDMP, (**d**) result of proposed method.

**Figure 10 sensors-22-08132-f010:**
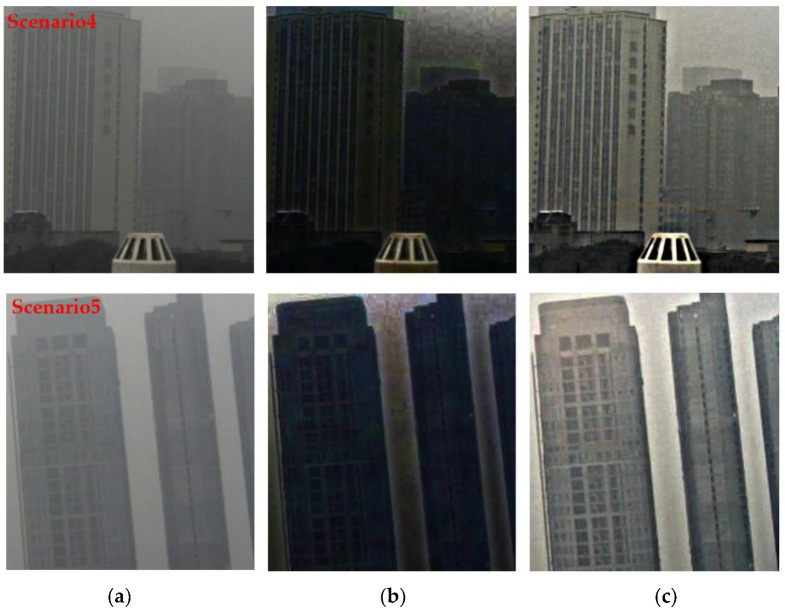
The comparison of deferent methods, (**a**) the original hazy image, (**b**) the result of DTMS, (**c**) the result of proposed method.

**Figure 11 sensors-22-08132-f011:**
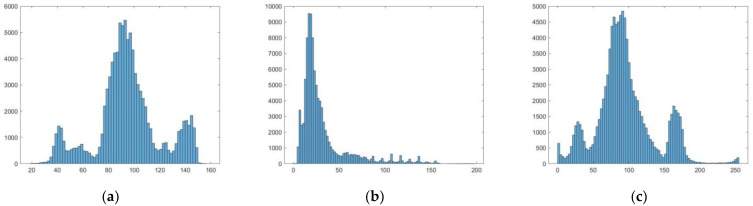
Histogram comparison results, (**a**) the result of original image, (**b**) the result of DTMS, (**c**) the result of proposed.

**Table 1 sensors-22-08132-t001:** Quantitative evaluation of defogging algorithms in the three scenes shown in Figure 7.

Grayscale Standard Deviation	Scenario1	Scenario2	Scenario3
Original	37.7078	56.9911	59.3694
Dark channel prior	68.4882	86.6204	84.1204
SIDMP	70.3881	83.5740	91.6790
Proposed	64.7314	77.6341	83.3641
Average Gradient	Scenario1	Scenario2	Scenario3
Original	0.0036	0.0045	0.0080
Dark channel	0.0050	0.0066	0.0084
SIDMP	0.0058	0.0067	0.0115
Proposed	0.0091	0.0096	0.0176

**Table 2 sensors-22-08132-t002:** Image quality evaluation without reference.

NIQE	Scenario4	Scenario5
Original	4.8812	5.3302
DTMS	5.1897	5.6805
Proposed	4.3335	4.2907

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
