# Peer review of "Defogging Algorithm Based on Polarization Characteristics and Atmospheric Transmission Model"

_sensors, 2022, doi:10.3390/s22218132_

Round 1

Reviewer 1 Report

see the attachment.

Author Response

Dear Editors and Reviewers:

Thank you for your letter and for the reviewers’ comments concerning our manuscript "Defogging algorithm based on polarization characteristics and atmospheric transmission model". These comments are all valuable and helpful for revising and improving our paper, as well as the important guiding significance to our researches. We have studied comments carefully and have made corrections which we hope meet with approval. The main corrections in the manuscript and the responds to the reviewer’s comments are as following:

  1. Please check the author’s name and other information are correct, such as Feng Ling, Zhang Yan.

Thank you for pointing this out. We are very sorry for our negligence. Authors' names and other information have been carefully checked and corrected.

  1. The introduction and related work of the manuscript need to be improved.

The introduction to this article has been carefully revised to highlight the advantages of this approach over other approaches and the applicable scenarios. The number of references has been increased to 34, and most of them came from the last past two years.

  1. There are many other defogging methods. How is this proposed method novel compared to theirs?

The dehazing comparison algorithm has been added and the motivation and details of the method design in this paper are supplemented.

  1. The cacography, fonts, and pictures in the flow chart are not clear and need to be carefully reviewed.

The flow chart has been redrawn and the text and pictures in it have been checked and confirmed.

  1. In formula (1), variable I is not explained.

Thank you for pointing this out. We are very sorry for our negligence, and variable I has been explained.

  1. The “considered in this study” written in the title of Figure 2 is redundant and suggested to be deleted.

The “considered in this study” written in the title of Figure 2 has been deleted.

  1. The word “detector” in Figure 2 is not right and is suggested to be corrected.

Figure 2 has been redrawn and the text images in the figure have been carefully examined.

  1. The meanings of subheadings (a), (b), and (c) in Figure 15 need to be explained.

The meanings of subheadings (a), (b), and (c) in Figure 15 has been checked and explained.

  1. The latest compared method here was published in 2011. Please also compare the proposed method with related approaches published in these two years.

Two experimental methods have been referenced for comparative tests, one of which was proposed in 2018, and the other in 2021.

  1. The formula numbering sequence needs to be corrected.

The formula numbering sequence has been checked carefully.

  1. All of the algorithms in Table 1 only need to introduce the name, the “algorithm” written behind each method is recommended to delete.

The “algorithm” written behind each method in table 1 have been deleted.

  1. This paper still lacks several ablation experiments to analyze the importance and effectiveness of each module utilized in the proposed framework.

In this paper, more experimental scenes are added to further demonstrate the experimental effect of this paper.

  1. The Conclusions must be improved.

The conclusion has been carefully examined and rewritten. In the conclusion, the reasons for the proposed method are emphasized, and the problems that the proposed method solved have been highlighted.

  1. If your results were unexpected, try to explain why. Is there another way to interpret your results?

In the conclusion experimental results obtained are compared with the previous hypotheses. The reasons for the problems in the results of this paper are analyzed, and the next work plan is put forward. The resulting image has been enlarged to show more details.

  1. The polarization method has achieved good results in image defogging. Like fog images, this method has also made good progress in underwater image processing. Related works need to be quoted.

The underwater image processing methods proposed in the list are closely related to the defogging methods, which strongly supports the writing of this paper and have been cited in the paper.

On behalf of my co-authors, we thank you very much for giving us an opportunity to revise our manuscript, we appreciate editor and reviewers very much for your positive and constructive comments and suggestions on our manuscript. I found the reviewer’s comments are quite helpful and valuable, and I revised my paper point-by-point. Thank you for your help.

Best Wishes

Feng Ling

Reviewer 2 Report

I’m not an expert on dehazing, but a simple google-search reveals several publications on the subject, presenting approaches that are not far from the one presented by the authors. In order to address this issue, the authors should start by providing a much larger reference list (preferably, including additional non-Chinese texts). The current number of bibliographic references seems small for a journal paper. The introduction should include the text to address those new references and it should clarify which presented parts are indeed novel and which parts come from published work. The introduction should also provide a brief piece of text mentioning the main applications or scenarios where image dehazing algorithms may potentially be useful. Other dehazing approaches, such those using deep learning, could also be mentioned.

The equations need to be strongly revised. Different fonts and font sizes are used along the manuscript, and notations and symbols aren't always handled with care. For instance, on section 2 it is difficult to clearly identify which symbols represent matrices/vectors and which ones represent scalars. It is also not clear if the convolution sign (*) is indeed convolution, or if it is used for multiplication.

Figures 3-a) to d) are not very helpful since they all look the same. The same goes for Figures 7a) to d) and 8-a) to d). The text addressing them is basically the corresponding cross-reference and does provide further insights. I suggest to include additional explanations / comments about the figures and to consider removing some of them in order to avoid redundancies.

Figure 5 depicts similar examples (a bit redundant, in my opinion) for the segmentation results, but no information on the parametrization for each case is provided. The differences should be highlighted, commented and the parametrizations should be depicted.

The comparison with other algorithms is somewhat poor since only three examples are provided. Also, I’m not entirely convinced that the presented quantitative metrics are enough for a fair comparison, since they are focused on the gradient only. Based on the examples provided, the proposed algorithm seems to produce darker and less colourful images in the non-sky regions when compared with the others, which may not be a good thing. The authors could assess quality using no-reference image quality metrics, for instance, and use a larger amount of examples.

The English language reads well. However, there are a few glitches that should be corrected in case of acceptance. The most obvious glitches are:
- a few spurious underlines seem to be forgotten along the text (for instance “an_infinite” in line 13, but are a few more along the document)
- figures 1 and 2 contain glitches on the in-figure text (“polaried”, “restore”, “Dtection”)
- the text mentions scene 1, scene 2, etc. but table 1 shows scene A, scene B, etc.
- line l33 - “2. Materials and Methods”

Author Response

Dear Editors and Reviewers:

Thank you for your letter and for the reviewers’ comments concerning our manuscript "Defogging algorithm based on polarization characteristics and atmospheric transmission model". These comments are all valuable and helpful for revising and improving our paper, as well as the important guiding significance to our researches. We have studied comments carefully and have made corrections which we hope meet with approval. Revised portion are marked in red in the manuscript. The main corrections in the manuscript and the responds to the reviewer’s comments are as following:

  1. The authors should provide a much larger reference list.

The number of bibliographic references has been increased to 34, and most of them came from the research in the past two years. In the introduction and conclusion, the innovative points and the applicable scenarios of this method are emphasized from other methods. What’s more, the dehazing approaches based on neural network are also introduced in the introduction.

  1. The equations need to be strongly revised.

Thank you for pointing this out. We are very sorry for our negligence. The formula and text have been strongly revised and unified in format.

  1. Figures 3-a) to d) are not very helpful since they all look the same.

Figures 3 has been deleted and in order to make the article look more concise, some redundant pictures and text have been processed according to your suggestions.

  1. Figure 5 depicts similar examples, the differences should be highlighted, commented and the parametrizations should be depicted.

Thank you very much for your kind suggestion, the segmentation of the sky region is the basis for defogging in this paper, so I consider to retain this part of the image and the differences between images have been highlighted in words.

  1. The authors could assess quality using no-reference image quality metrics, for instance, and use a larger amount of examples.

In order to better demonstrate the results of the proposed method, the results obtained in different experimental scenarios have been supplemented, and the reference-free image quality assessment method has been used to analyze the experimental results.

  1. There are a few glitches that should be corrected in case of acceptance.

Thank you for pointing this out. We are very sorry for our negligence. We have checked the writing of the article carefully and corrected the mistakes.

On behalf of my co-authors, we thank you very much for giving us an opportunity to revise our manuscript, we appreciate editor and reviewers very much for your positive and constructive comments and suggestions on our manuscript. I found the reviewer’s comments are quite helpful and valuable, and I revised my paper point-by-point. Thank you for your help.

Best Wishes

Feng Ling

Round 2

Reviewer 1 Report

The authors have revised this manuscript as the reviewers suggested, and I am satisfied with this new version. Therefore, I suggest this paper could be accepted for the possible publication.

Author Response

2022.10.4

Dear Editors and Reviewers:

Thank you for your letter and for the reviewers’ comments concerning our manuscript "Defogging algorithm based on polarization characteristics and atmospheric transmission model".

Thank you for your positive comments on this article, we will continue to work in this direction and strive for better results in the near future.

Wish you all the best in your work and life.

Best Regards

Feng Ling

Reviewer 2 Report

The new version of the paper is noticeably improved when compared with the first one. Most of my previous concerns have been addressed.

There are a few issues remaining:

- the equations layout requires additional work - for instance, eq. (14) looks rather ugly;
- fig. 6 is missing;
- "method2" and "method3" should be renamed to suitable designations (both on text and tables);
- there are minor glitches here and there (typically white spaces missing or spurious white spaces), but I guess the editing team can clear them.

Author Response

Dear Editors and Reviewers:

Thank you for your letter and for the reviewers’ comments concerning our manuscript "Defogging algorithm based on polarization characteristics and atmospheric transmission model". These comments are all valuable and helpful for revising and improving our paper, as well as the important guiding significance to our researches. We have studied comments carefully and have made corrections which we hope meet with approval.

  1. The equations layout requires additional work - for instance, eq. (14) looks rather ugly

We have made changes to the formula and explained the meaning of the parameters in the formula.

  1. 6 is missing

Thank you for pointing this out. We are very sorry for our negligence. There was an error converting a Word document to a PDF file, We have checked carefully this time to make sure it is correct.

  1. "Method2" and "Method3" should be renamed to suitable designations (both on text and tables)

Thank you very much for your kind suggestion, We have made changes to the names.

On behalf of my co-authors, we appreciate editor and reviewers very much for your positive and constructive comments and suggestions on our manuscript. I found the reviewer’s comments are quite helpful and valuable, and I revised my paper point-by-point. Thank you for your help.

Best Wishes

Feng Ling